# PCP: A Pseudonym Change Scheme for Location Privacy Preserving in VANETs

**DOI:** 10.3390/e24050648

**Published:** 2022-05-05

**Authors:** Xinyang Deng, Tianhan Gao, Nan Guo, Cong Zhao, Jiayu Qi

**Affiliations:** 1Software College, Northeastern University, Shenyang 110819, China; xinyang1121@sina.com (X.D.); gaoth@mail.neu.edu.cn (T.G.); cc598487409@gmail.com (C.Z.); yoonagoogoo@gmail.com (J.Q.); 2Computer Science and Engineering College, Northeastern University, Shenyang 110819, China

**Keywords:** VANETs, pseudonym change, location privacy

## Abstract

In vehicular ad hoc networks (VANETs), pseudonym change is considered as the vital mechanism to support vehicles’ anonymity. Due to the complicated road conditions and network environment, it is a challenge to design an efficient and adaptive pseudonym change protocol. In this paper, a pseudonym change protocol for location privacy preserving (PCP) is proposed. We first present the requirements of pseudonym change in different scenarios. According to variable network states and road conditions, vehicles are able to take different pseudonym change strategies to resist the tracking by global passive adversaries. Furthermore, the registration protocol, authentication protocol, pseudonym issuance protocol, and pseudonym revocation protocol are introduced for the pseudonym management mechanism. As a consequence, it is not feasible for global passive adversaries to track a vehicle for a long time and obtain the trajectory of the vehicle. The analysis results show that the security and performance of PCP are improved compared with the traditional ones.

## 1. Introduction

The intelligent transportation system (ITS) is regarded as an important part of next-generation urban transport, which integrates a variety of advanced technologies (e.g., sensor technology, intelligent control technology) to improve convenience for drivers and pedestrians [1]. Being able to keep a stable network connection and provide a diversity of services, vehicular ad hoc networks (VANETs), as the essential part of the ITS, have had increasing attention paid to them [2]. According to the current features of urban traffic (e.g., rapid vehicle movement, uneven traffic distribution), VANETs have formed the standard in line with the future development of intelligent transportation [3], while still facing the following challenges: (1) Fast topology change: The fast-changing network topology caused by the instability of vehicle velocity has put forward more requirements for VANETs to provide stable network communication services, such as routing algorithms and congestion prediction mechanisms. (2) Non-static network density: The fast topology change causes the continuous change of the service intensity of roadside units (RSUs), which leads to a delay in responding to requests from vehicles. In addition, the instability of the signal-to-noise ratio caused by network density also affects the stability of communication. (3) Wireless communication environment: Owing to the wireless medium’s nature, it is difficult to protect the security of communication. (4) Limited communication duration: Vehicles need to avoid performing high calculations or storing excessive data to complete the authentication and data transmission as soon as possible.

Figure 1 shows the framework of VANETs. RSUs, as the roadside infrastructure, are deployed on both sides of the road. RSUs are able to collect the driving state of the surrounding vehicles, predict the traffic flow situation nearby, provide certain driving suggestions for vehicles, and support the road condition warning service. In addition, RSUs support providing network services for vehicles by connecting with the base station. Vehicles equipped with on-board units (OBUs) can communicate with surrounding vehicles and RSUs to obtain a variety of application services. In addition, each vehicle is also secured with a GPS receiver to have an accurate location and time [4].

In order to ensure the driving safety of vehicles, vehicles are required to send the message related to their driving status regularly to surrounding vehicles and RSUs [5], e.g., basic safety message (BSM) [6]. The BSM guarantees that vehicles are aware of the danger so as to make appropriate decisions in time. However, the adversaries in the communication range are able to collect and aggregate received data through eavesdropping on the BSM. Consequently, the location privacy of vehicles and the individual privacy of vehicles owners are threatened. For the purpose of protecting vehicle location privacy, the IEEE 1609.2 standard suggests using a pseudonym instead of the real identity [7]. Accordingly, it becomes impossible to obtain the private information of the vehicle owners through utilizing the real identity of vehicles. However, if there is no effective strategy to support pseudonym change, the adversaries can still link the pseudonym and the real identity through tracking vehicles for a long time, thus invading the location privacy of vehicles [8].

Figure 2 and Figure 3 depict the syntactic linking scenario and the semantic linking scenario of VANETs, respectively [9].

In the syntactic linking scenario, if a vehicle changes from pseudonym $PS_{1}$ to pseudonym $PS_{2}$ while other vehicles decide not to change their pseudonyms at the same time, it is obvious that the adversaries are able to determine that $PS_{1}$ and $PS_{2}$ are from the same vehicle. In the semantic linking scenario, vehicles at the intersections are required to change their pseudonyms. However, if the vehicle (holding $PS_{1}$) does not change its trajectory or there is no vehicle with a similar driving status around the vehicle, the adversaries are still able to utilize advanced tracking algorithms to predict the location of the vehicle according to the BSM regularly sent by the vehicle. As a result, the semantic linking attack makes adversaries believe that $PS_{1}$ and $PS_{2}$ belong to the same vehicle.

Moreover, the frequency of pseudonym change is the important factor that affects the location privacy degree. The higher the frequency of pseudonym change, the better the degree of privacy protection. However, due to limited bandwidth, the frequency should not seriously hinder the performance. Consequently, it is crucial to design a secure and efficient pseudonym change scheme to guarantee that any adversaries can not associate the same vehicle with two different pseudonyms or track a certain vehicle for a long time.

Up to the present moment, a large number of pseudonym change strategies have been proposed, such as mix zones and silent periods. The core idea of these strategies is to find or create opportunities to break the continuous tracking of vehicles. However, in the silent period mechanism, the time window is limited by the interval of the BSMs. In the mix zone mechanism, the security of the pseudonym change depends on the number of pseudonyms changed synchronously in the mix zone. Moreover, the glaring issue is that the strength of the privacy protection of the above schemes heavily depends on the vehicle density within the communication range. Under low-density conditions, it is difficult to keep high location privacy. In order to address the above issues, we propose a novel pseudonym change scheme for location privacy preserving in VANETs (PCP):(1)We improve the ID-based linearly homomorphic signature scheme and construct a pseudonym generation and aggregate protocol, where vehicles are able to calculate legitimate pseudonym certificates without the participation of the RSUs. Meanwhile, vehicles can judge the conditions for pseudonym change independently and obtain the necessary information through vehicle-to-vehicle (V2V) communication to enhance the safety of the subsequent pseudonym change protocol.(2)The vehicle registration protocol, authentication protocol, and pseudonym revocation protocol are proposed, which guarantee that all legal vehicles are able to communicate with surrounding entities and compromised vehicles can be revealed in time.(3)The computational cost and communication cost are adopted to evaluate the performance of the V2I authentication protocol in PCP. In addition, the vehicles in network simulation framework (Veins) is introduced to simulate the pseudonym change protocol of the proposed scheme to verify the effectiveness.

The remainder of the paper is organized as follows. Section 2 discusses the related works on pseudonym change. Section 3 revisits the preliminaries and presents the improved identity-based signature mechanism. The details of the proposed scheme (PCP) are given in Section 4. Section 5 and Section 6 analyze the security and performance of PCP, respectively. Finally, we conclude the work and present the future work in Section 7.

## 2. Related Works

In recent years, a large number of pseudonym change schemes have been proposed. Generally, the taxonomy of pseudonym change strategies includes mix-zone-based strategies and silent-period-based strategies.

### 2.1. Mix-Zone-Based Strategies

This strategy requires that vehicles change their pseudonyms in fixed areas, called mix zones, where the location of the mix zones is usually determined by the RSUs. Ref. [10] proposed cryptographic mix zones (CMIX zones). In a CMIX zone, the BSMs are transmitted as ciphertext. External adversaries cannot obtain any useful information related to the pseudonym change. However, the proposed scheme does not consider the size of the anonymity set. If there are few vehicles in a CMIX zone, adversaries still have the ability to track the target vehicle with high probability. In order to solve the problem, Lu et al. suggested a mix zone that is deployed at social spots [11], such as intersections or a spot near a shopping mall. The most feasible case is the intersection with high traffic flow and traffic lights, where there are a large number of slow-moving vehicles that have enough time to change their pseudonyms. Refs. [12,13,14] utilized roadside infrastructure to support vehicle pseudonym changing. Ref. [12] suggested building the vehicular location privacy zone, where two infrastructures called the router and aggregator are deployed at both ends of the vehicular location privacy zone (VLPZ), which are responsible for ensuring the unlinkability of the pseudonym changing, respectively. When a vehicle arrives at the router in the VLPZ, the vehicle stops broadcasting the BSM. The router selects a lane for the vehicle randomly, and the vehicle is required to change its pseudonym before reaching the aggregator. As the exit order is different from the entering order due to random residency periods, it is difficult to link the same vehicle. Ref. [13] depended on fog computing to provide the pseudonym change service for vehicles. Different from ref. [12], the new pseudonyms for all vehicles in the mix zone are provided by the RSUs. Ref. [13] alleviated the computational cost and storage cost of the central authority to improve the efficiency of updating pseudonyms. In the above schemes, the shared key is usually adopted to resist external attacks. However, if a vehicle is compromised, the adversaries can eavesdrop on the communication message from the vehicle inside the mix zone and still be able to track the target vehicle. In order to solve this issue, ref. [14] proposed a pseudonym swap scheme based on differential privacy. When a vehicle needs to change its pseudonym, the vehicle sends the request message to the RSUs and the surrounding vehicles. Other vehicles that need to change the pseudonym send the same message to the RSUs and apply to join the pseudonym swap. The RSUs collect the request messages and use the pseudonym swap algorithm to assign a new pseudonym for each vehicle according to differential privacy. The scheme realizes a pseudonym exchange scheme where the RSUs have the ability to guarantee pseudonym indistinguishability and achieve the unlinkability between the new pseudonym and the old one. However, the heavy computation and communication costs result in the low efficiency of the scheme. In LIAP [15], vehicles are able to use the certificate from the CA to enter the security domain in the RSU. The RSU is required to periodically change the public key in the domain, and vehicles can change their pseudonyms according to the change of the public key, thereby guaranteeing that the pseudonyms are changed periodically. Nevertheless, vehicles have to communicate with the RSU before changing their pseudonyms. As a result, under special conditions, the pseudonym of vehicles cannot be changed since they cannot communicate with the RSU in time.

### 2.2. Silent-Period-Based Strategies

The silent-period-based strategy refers to the transition period of the pseudonym change. In a silent period, no vehicle is allowed to disclose either the old or the new identity and location [16]. Different from mix-zone-based strategies, silent-period-based strategies support the vehicles in choosing the area of the pseudonym change independently, and the time of the pseudonym change can be determined through negotiation among the vehicles. Normally, silent-period-based strategies require that the vehicles in VANETs establish a group through communication. These vehicles in the group determine the time and mechanism of the pseudonym change, and other vehicles outside the group cannot obtain any useful information within the group [17]. In ref. [18], vehicles detected whether the surrounding vehicles have the possibility of expected the cooperation in the pseudonym exchange by receiving the BSM. If the driving state of the surrounding vehicles is similar to the vehicle, the pseudonym change scheme will be activated. When changing the pseudonym, each vehicle is requested to broadcast a BSM with the position where the pseudonym change starts and set the speed to 0 until the pseudonym change is complete. Nevertheless, since vehicles cannot provide accurate road information to the owners, a serious impact on traffic may be caused consequently. In ref. [19], each vehicle owned a time-slotted pseudonym pool. In each time slot, only one pseudonym is legal. At the end of each time slot, vehicles are required to exchange pseudonyms to guarantee anonymity. In particular, the time of exchanging pseudonyms is determined by the driving state of the surrounding vehicles. The proposed scheme eliminates the mapping between the pseudonym and vehicle and achieves the reuse of old pseudonyms. Furthermore, due to the fixed-size pseudonym pool, the workload of the certificate authority (CA) only depends on the number of vehicles joining the network. However, the scheme does not give the details of to verify the legality of the pseudonym. Ref. [20] provided a pseudonym changing strategy (SLOW), which does not require extensive RSUs or a complex communication procedure. When the speed of the vehicle is slower than the given threshold, the vehicle stops broadcasting the BSM and any other message containing location or trajectory data and changes its pseudonym. However, if the vehicle stops broadcasting the BSM, it is difficult for other vehicles to accurately obtain the surrounding road condition information [21]. Ref. [22] proposed a cooperative pseudonym change scheme based on a trigger. In the proposed scheme, a “Readyflag” bit is inserted into the BSM. According to the value of “Readyflag” (0 or 1) in the received BSM, the vehicle determines whether to cooperate with the vehicles in the vicinity to change the pseudonym together. Ref. [22] not only enabled vehicles to obtain the willingness of surrounding vehicles to change pseudonyms in time, but also expanded the size of the anonymous set. However, ref. [22] did not give the details about how to change the pseudonym. Besides, the influence of the vehicle running state on the security of pseudonym change was not considered. Ref. [23] proposed Mix Group to solve the issue that a small group is weak in preserving privacy while a large-scale group leads to low efficiency in managing the signatures. According to the “Pareto principle”, Mix Group supports the pseudonym exchange protocol for vehicles with a common driving status under any road condition, which guarantees that the location privacy is substantially enhanced. However, the pseudonym exchange is carried out independently between vehicles. Once a vehicles is compromised, it is difficult to track the illegal vehicle. Ref. [24] gave three options: cooperative pseudonym exchange (CPE), scheme permutation (SP), and CPE plus SP (CPESP), to improve the location privacy. Vehicles are able to choose the appropriate option according to different traffic statuses. As the scheme does not give the details about the pseudonym change, we cannot determine the security of the scheme. In SPA [25], vehicles store the password issued by the TA in tamper-proof devices (TPDs). The TDP is responsible for generating and changing the pseudonyms of vehicles. However, in order to protect the privacy, vehicles have to choose the appropriate time to change the pseudonym according to the nearby road conditions. Ref. [26] adopted blockchain technology to support the location privacy preserving of vehicles (BELP). BELP removes the central authority, which effectively prevents vehicle privacy from being tampered with or leaked by internal adversaries. However, the proposed scheme does not give the details of pseudonym generation and illegal vehicle revocation. Once a vehicle misbehaves, it is critical to track the vehicle and remove it from the VANET in a timely manner.

Due to the heavy dependence on the deployment density of mix zones and the driving state of surrounding vehicles, mix-zone-based strategies lack the flexibility to support the pseudonym change. Silent-period-based strategies makes the vehicle unable to transmit or receive accurate road condition information in time, which may affect the driving safety of the vehicle. Consequently, it is very important to design an effective mechanism to adapt to the pseudonym change in various scenarios.

## 3. Preliminaries

### 3.1. Bilinear Pairing

Let $G_{1}$ be the additive cyclic group of prime order *q* with λ bits and $G_{T}$ be the multiplicative cyclic group of the same order. $e:G_{1}\times G_{1}\to G_{T}$ is a bilinear pairing with the following properties [27]:

(1) Bilinearity: $\forall P,Q,\leftarrow G_{1}$ and $\forall a,b\leftarrow Z_{q}^{*}$, there is $e\left(P^{a},Q^{b}\right)=e{\left(P,Q\right)}^{ab}$.

(2) Non-degeneracy: $\exists P,Q\leftarrow G_{1}$, $e\left(P,Q\right)\neq 1_{G_{T}}$.

(3) Computability: $\forall P,Q\leftarrow G_{1}$, there exists an efficient algorithm to calculate e(P,Q).

### 3.2. Computational Diffie–Hellman Assumption

Given a random generator $P\leftarrow G_{1}$, random numbers *a*, *b*$\leftarrow Z_{q}^{*}$, and security parameter λ, the advantage of an algorithm *A* in solving the computational Diffie–Hellman problem in group $G_{1}$ is
$$\begin{array}{c} ADV_{A}^{CDH}\left(\lambda \right)=Pr\left[A\left(P,aP,bP\right)=abP\right] \end{array}$$

We say that an algorithm A(t,τ)-breaks the computational Diffie–Hellman problem in $G_{1}$ if *A* runs in time at most *t* and $ADV_{A}^{CDH}\geq \tau$.

### 3.3. Identity-Based Signature Mechanism

The identity-based signature (IBS) is a special signature where the verifier is able to verify the signature given the identity information from the signer. PCP adopts the CC signature [28] and improves Lin’s signature scheme [29] to support the anonymous authentication protocol and pseudonym change protocol, which includes a tuple of four PPT algorithms: Setup, Extract, Sign, Verify.

$\left(msk,params\right)\leftarrow Setup\left(1^{\lambda }\right)$. Let $G_{1}$, $G_{T}$ be the additive group and multiplicative group such that $|G_{1}\left|=\right|G_{T}|=q$. A bilinear pairing is defined by $e:G_{1}\times G_{1}\to G_{T}$. Given three hash functions $H:{\left\{0,1\right\}}^{*}\to Z_{q}^{*}$, $H_{1}:{\left\{0,1\right\}}^{*}\to G_{1}$, $H_{2}:{\left\{0,1\right\}}^{*}\to G_{1}$, generator $P\leftarrow G_{1}$, and random number $x^{\prime }\leftarrow Z_{q}^{*}$, compute $\left(x,P_{pub}\right)\leftarrow CC\_Setup\left(1^{\lambda }\right)$, $P_{pub}^{\prime }=x^{\prime }P$. $Setup(1^{\lambda })$ outputs the master key $msk=\{x,x^{\prime }\}$ and the public parameters $params=\{G_{1},G_{T},q,e,P,P_{pub},P_{pub}^{\prime },H,H_{1},H_{2}\}$.

$SK_{ID}\leftarrow Extract\left(msk,ID\right)$. Given the master key msk and user identity ID, return secret key $SK_{ID}=\left\{SK,SK^{\prime }\right\}$ corresponding to ID, where SK=CC_Extract(x,ID), $SK^{\prime }=x^{\prime }H_{1}\left(ID\right)$.

$\sigma \leftarrow Sign(ID,SK_{ID},M_{1},M_{2})$. This algorithm takes the user identity ID, secret key $SK_{ID}$, and messages $M_{1}$, $M_{2}$ as the input and outputs the signature $\sigma =\{\sigma_{1},\sigma_{2},w,s\}$, where $\sigma_{1}=CC\_Sign\left(w\right)$, $\sigma_{2}=H\left(M_{1}\right)SK^{\prime }+r\left(sH_{1}\left(ID\right)+H\left(M_{1}\right)H_{2}\left(M_{2}\right)\right)$, w=rP, $r,s\leftarrow Z_{q}^{*}$.

{0or$1\}\leftarrow Verify\left(ID,M_{1},M_{2},\sigma \right)$. Given the signer’s identity ID, messages $M_{1},M_{2}$, and signature σ, the verifier checks $CC\_Verify\left(ID,w,\sigma_{1}\right)\overset{?}{=}1$ and $e\left(\sigma_{2},P\right)\overset{?}{=}e(H_{1}\left(ID\right)$, $P_{pub}^{\prime }{)}^{H(M_{1})}e\left(sH_{1}\left(ID\right)+H\left(M_{1}\right)H_{2}\left(M_{2}\right),w\right)$. If both of the above equations hold, output 1, and 0 otherwise.

The security of the above signature algorithms is based on the CDH assumption. The formal security proof is detailed in Section A.2.

## 4. The Proposed Scheme

In this section, a pseudonym change scheme for location privacy preserving in VANETs is elaborated. Figure 4 shows the scenario and participating entities of each protocol, which include system initialization, the registration protocol, the authentication protocol, the pseudonym issuance protocol, the pseudonym change protocol, and the pseudonym revocation protocol. In addition, the system architecture, adversary model, and security requirements are introduced first before describing the details of the scheme. The notations and descriptions are listed in Table 1.

### 4.1. System Architecture

Figure 5 shows the system architecture of PCP, which includes four components: trust authority (TA), base station (BS), roadside unit (RSU), and vehicle.

The TA is responsible for generating public parameters, pseudonyms, and public/private key pairs for vehicles, BSs, and RSUs. In addition, when a vehicle is compromised or conducts illegal behavior, the TA can assist other entities to disclose the real identity of the vehicle and exclude the vehicle from the system in time.

The BSs are deployed in multiple regions in the city, and the RSUs in each region are managed by the BSs. Besides, in PCP, the BSs generate temporary pseudonyms for vehicles in the region.

The RSUs adopt DSRC/WAVE to connect with the vehicles in the vicinity and provide a series of application services for legal vehicles [3,30]. Meanwhile, RSUs provide the pseudonyms’ related authentication and change services for the vehicles.

The vehicles communicate with the surrounding RSUs and other vehicles to obtain the services. In order to protect vehicles’ location privacy, available strategies are required to support pseudonym changes under the four scenarios shown in Figure 4.
**Scenario 1**: In the area with a low vehicle density and no RSUs, if there are non-negligible differences in vehicle driving statuses, it is difficult to make an effective mechanism of pseudonym change in order to resist the tracking of external attackers. However, we hope to provide an efficient mechanism to make full use of such a scenario and obtain enough useful information as much as possible, so as to provide a higher level of location privacy preserving.**Scenario 2**: There is a high vehicle density in this area, and RSUs exist to provide services for surrounding vehicles. In this scenario, the vehicles and RSUs can cooperate to change their pseudonyms and resist the attacks from external adversaries for protecting the location privacy of vehicles.**Scenario 3**: The RSUs exist, but the vehicle density is low. The RSUs can provide the pseudonym update service for vehicles that are running out of pseudonyms. Multiple pseudonym change mechanisms are available.**Scenario 4**: This area has a high vehicle density without RSUs. The vehicles can use the pseudonym change mechanism to change their pseudonyms through their cooperation.

Since a variety of pseudonym change schemes are proposed in Scenario 2 and Scenario 3, PCP focuses on the details of pseudonym change in Scenario 1 and Scenario 4.

### 4.2. Adversary Model

It is assumed that the adversaries are the global passive adversaries (GPAs). The global adversary holds the capacity to eavesdrop on the communication message of the whole network. The passive adversary refers to the adversary that does no more than eavesdropping on the communication traffic in the VANET [9]. Therefore, a GPA has the ability to eavesdrop on the BSMs of all vehicles in the region of interest. In PCP, we assume that the GPAs know the pseudonym change strategy and vehicles are required to broadcast BSMs to vehicles in the vicinity periodically while driving, which includes the identifier, position, velocity, direction, etc. If a vehicle does not change its identifiers for a long time, the GPAs are able to eavesdrop on the BSM sent by the vehicle, track the designated vehicle, and obtain the vehicle’s trajectory and privacy via the syntactic linking attack and the semantic linking attack.

### 4.3. Security Requirements

In this section, we assume that the TA is honest and trustworthy, but there is no trust relationship among the other entities in the VANET. According to [8,31], the proposed scheme should meet the following goals:Anonymity: No adversary is able to extract the vehicle’s real identity from its pseudonym. The identities broadcast by vehicles are required to be anonymous within a set of potential vehicles, which ensures that no entities can obtain useful information about the real identity of vehicles. Moreover, anonymity is supposed to be conditional according to the security requirements of VANETs.Unlinkability: If the adversaries can obtain the messages sent by vehicles through monitoring, it is difficult to determine whether the consecutive received messages are sent by the same vehicle. In the pseudonym change protocol, no pseudonym should reveal any connections among vehicles.Mutual authentication: As the basic security requirement, mutual authentication focuses on identities and messages. Identity authentication means that the identity claimed by the entity is legal. Message authentication requires that the integrity of the message be able to be verified.Traceability: In a secure network architecture, it is essential to provide an efficient mechanism to trace the origin of the message. However, such a mechanism can only be effective under an authorized authority.Session key agreement: For data transmission, the confidentiality of the data is also a security requirement of VANETs. Therefore, after finishing the initial authentication, designing a session key agreement mechanism between entities in VANETs to encrypt the communication messages usually needs to be considered.Location privacy: Vehicle owners usually do not want their location to be exposed in sensitive areas. Consequently, vehicles need to change their identity information at specific areas, so that the adversaries cannot track the specific vehicle for a long time or obtain the driving trajectory.DoS attack resistance: The external adversaries are able to forge and broadcast a large number of invalid messages to consume the computational resource of the vehicles, which leads to legitimate messages possibly being dropped. As a result, it is necessary to ensure a low computational overhead for vehicles during communication.

### 4.4. System Initialization

During system initialization, the TA generates and broadcasts public parameters to the whole network. The details are shown as follows:Let $G_{1}$ and $G_{T}$ be the additive group and multiplicative group, respectively, where $|G_{1}\left|=\right|G_{T}|=q$ for the same prime order *p*. *P* is the generator of $G_{1}$. Let *e* be a bilinear pairing: $G_{1}\times G_{1}\to G_{T}$.Six collision-resistant hash functions are defined: $H:{\left\{0,1\right\}}^{*}\to Z_{q}^{*}$, $H_{1}:{\left\{0,1\right\}}^{*}\to G_{1}$, $H_{2}:{\left\{0,1\right\}}^{*}\to G_{1}$, $H_{3}:{\left\{0,1\right\}}^{*}\times Z_{q}^{*}\to Z_{q}^{*}$, $H_{4}:{\left\{0,1\right\}}^{*}\times {\left\{0,1\right\}}^{*}\to G_{1}$, $H_{5}:{\left\{0,1\right\}}^{*}\times G_{1}\to Z_{q}^{*}$.The TA chooses $x,x^{\prime }\leftarrow Z_{q}^{*}$ as the master key and $s\leftarrow {\left\{0,1\right\}}^{n}$ as the key of the AES-256 encryption algorithm and computes the public key $P_{pub}=xP$, $P_{pub}^{\prime }=x^{\prime }P$.

The TA broadcasts public parameters param=$\{G_{1}$, $G_{T}$, *q*, *e*, *P*, $P_{pub}$, $P_{pub}^{\prime }$, *H*, $H_{1}$, $H_{2}$, $H_{3}$, $H_{4}$, $H_{5}\}$ to all entities in the system.

### 4.5. Registration Protocol

#### 4.5.1. Vehicle Registration Protocol

When vehicle *v* with $ID_{v}$ enters the VANET, it requests to apply for registration with the TAs. The TAs are able to generate a series of pseudonyms ${\left\{PS_{i}\right\}}_{i\in [1,n]}$, public keys ${\left\{PK_{i}\right\}}_{i\in [1,n]}$, and private keys ${\left\{SK_{i}\right\}}_{i\in [1,n]}$ for the vehicle. The protocol is performed as Figure 6 and Protocol 1.
The vehicle chooses session key $K_{v-TA}\leftarrow {\left\{0,1\right\}}^{n}$ and encrypts $K_{v-TA}$ and $ID_{v}$ to obtain $C_{v-TA}=Enc\_P_{pub}\{ID_{v}$, $K_{v-TA}\}$. Then, the vehicle sends $C_{v-TA}$ to the TA for registration.The TA uses *x* to decrypt $C_{v-TA}$ to obtain $ID_{v}$, $K_{v-TA}$. Then, the TA chooses ${\left\{x_{i}\right\}}_{i\in [1,n]}\leftarrow$$Z_{q}^{*}$ and computes the corresponding pseudonyms $PS=\{PS_{i}|i\in \left[1,n\right]\}$, public keys $PK=\{PK_{i}|i\in \left[1,n\right]\}$, private keys $SK=\{SK_{i}|i\in \left[1,n\right]\}$, and the expiration $EXP=\{EXP_{i}|i\in \left[1,n\right]\}$, where $PS_{i}=Enc\_s\{ID_{v}\left|\right|x_{i}\}$, $PK_{i}=H_{1}(PS_{i}||EXP_{i})$, $SK_{i}=xPK_{i}$.The TA utilizes $K_{v-TA}$ to encrypt PS, EXP, and SK and obtains $C_{TA-v}$ = $Enc\_K_{v-TA}\left\{PS||EXP||SK\right\}$.Upon receiving the message from the TA, vehicle $v_{i}$ uses $K_{v-TA}$ to decrypt $C_{TA-v}$ to obtain PS, EXP, and SK.

#### 4.5.2. BS and RSU Registration Protocol

In this protocol, the BS is able to obtain its public key $PK_{BS}$, private key $SK_{BS},SK_{BS}^{\prime }$, and expiration $EXP_{BS}$, and the RSU can obtain its public/private key $PK_{RSU}/SK_{RSU}$, and expiration $EXP_{RSU}$ from the TA via a secure channel, where
$$\begin{array}{cc} PK_{BS} & =H_{1}(ID_{BS}||EXP_{BS}),\\ SK_{BS} & =xPK_{BS},\\ SK_{BS}^{\prime } & =x^{\prime }PK_{BS},\\ PK_{RSU} & =H_{1}(ID_{RSU}||EXP_{RSU}),\\ SK_{RSU} & =xPK_{RSU}. \end{array}$$
Finally, the BS chooses $r_{BS}\leftarrow Z_{q}^{*}$ and computes the public key $P_{pub}^{BS}=r_{BS}P$ used in the BS domain.
**Protocol 1** Vehicle registration protocol.1:*v*:   choose $K_{v-TA}\leftarrow {\left\{0,1\right\}}^{n}$;   encrypt $K_{v-TA}$ and $ID_{i}$ to obtain $C_{v-TA}=Enc\_P_{pub}\{ID_{v}$, $K_{v-TA}\}$;2:v→TA: $C_{v-TA}$;3:TA:   decrypt $C_{v-TA}$ to obtain $ID_{v}$, $K_{v-TA}$;   choose $x_{1}$, $x_{2}$, …, $x_{n}$←$Z_{q}^{*}$   compute $PS=\{PS_{i}|i\in \left[1,n\right]\}$, $PK=\{PK_{i}|i\in \left[1,n\right]\}$, $SK=\{SK_{i}|i\in \left[1,n\right]\}$, where      $PS_{i}=Enc_{s}\{ID_{v}\left|\right|x_{i}\}$      $PK_{i}=H_{1}(PS_{i}||EXP_{i})$      $SK_{i}=xPK_{i}$   compute $C_{TA-v}=Enc\_K_{v-TA}\left\{PS||EXP||SK\right\}$;4:$TA\to v_{i}$:    $C_{TA-v}$;5:*v*:   decrypt $C_{TA-v}$ to obtain PS, EXP, SK   store PS, EXP, SK locally.

### 4.6. V2I Authentication and Pseudonym Issuance Protocols

When entering the signal coverage of the RSU, the vehicle is able to apply for new pseudonyms from the BS via the RSU. The RSU first verifies the legality of the vehicle through V2I authentication. If the vehicle is legal, the BS issues multiple pseudonyms for the vehicle, where these pseudonyms are valid within the scope of the BS.

#### 4.6.1. V2I Authentication Protocol

V2I authentication supports the establishment of the trust relationship between the vehicle and RSU, as well as the construction of a secure channel. The details are depicted in Figure 7 and Protocol 2.
Vehicle *v* chooses $PS_{i}$, $SK_{i}$, and $EXP_{i}$ and signs message $PS_{i}$, $EXP_{i}$, $TS_{1}$, $N_{1}$, and $r_{v}P$ to obtain signature $sign_{v}=\left\{V,W\right\}$, where $r_{v}\in Z_{q}^{*}$, $V=r_{v}H_{1}(PS_{i}||EXP_{i})$, $h=H_{5}(PS_{i}||EXP_{i}\left|\right|$$TS_{1}\left|\right|N_{1}\left|\right|r_{v}P,V)$, $W=\left(r_{v}+h\right)SK_{i}$.The vehicle sends $PS_{i}$, $EXP_{i}$, $TS_{1}$, $N_{1}$, $r_{v}P$, and $sign_{v}$ to the RSU.When receiving the message from the nearby *v*, the RSU first checks whether $TS_{1}$ and $EXP_{i}$ are fresh. Then, the RSU computes *h*=$H_{5}(PS_{i}||EXP_{i}||TS_{1}\left|\right|N_{1}\left|\right|r_{v}P,V)$ and $PK_{i}$=$H_{1}(PS_{i}\left|\right|$$EXP_{i})$. After that, the RSU checks whether $e\left(P,W\right)=e\left(P_{pub},V+hPK_{i}\right)$ holds. If the above equations are valid, the RSU believes *v* is legal. Otherwise, the message from the vehicle is discarded. The RSU signs $ID_{RSU}$, $EXP_{RSU}$, $TS_{2}$, $N_{2}$, and $r_{RSU}P$ to obtain $sign_{RSU}=\left\{V^{\prime },W^{\prime }\right\}$, where $r_{RSU}$∈$Z_{q}^{*}$, $V^{\prime }$=$r_{RSU}H_{1}(ID_{RSU}||EXP_{RSU})$, $h^{\prime }$=$H_{5}(ID_{RSU}||EXP_{RSU}||TS_{2}\left|\right|N_{2}\left|\right|$$r_{RSU}P,V^{\prime })$, $W^{\prime }$ = $\left(r_{RSU}+h^{\prime }\right)SK_{RSU}$. Finally, the RSU computes session key $K_{RSU-v}$=$r_{RSU}r_{v}P$ and encrypts $N_{1}$ to obtain $C_{RSU-v}$=$Enc\_K_{RSU-v}\left\{N_{1}\right\}$.The RSU sends $ID_{RSU}$, $EXP_{RSU}$, $TS_{2}$, $N_{2}$, $r_{RSU}P$,$sign_{RSU}$, and $C_{RSU-v}$ to *v*.Upon receiving the message from the RSU, *v* checks $TS_{2}$, $EXP_{RSU}$ and verifies the legality of $sign_{RSU}$ through computing $PK_{RSU}$, $h^{\prime }$, and checking $e\left(P,W^{\prime }\right)\overset{?}{=}e\left(P_{pub},V^{\prime }+h^{\prime }PK_{RSU}\right)$. If the equation holds, $v_{i}$ computes $K_{v-RSU}=r_{v}r_{RSU}P$ and decrypts $C_{RSU-v}$ to obtain $N_{1}$. If $N_{1}$ is legal, $v_{i}$ believes RSU is legal, and the secure channel between $v_{i}$ and the RSU is established. Finally, *v* encrypts $N_{2}$ to obtain $C_{v-RSU}=Enc\_K_{v-RSU}\left\{N_{2}\right\}$.Vehicle *v* sends $C_{v-RSU}$ to the RSU.The RSU decrypts $C_{v-RSU}$ and checks $N_{2}$. If $N_{2}$ is valid, the RSU believes that the secure channel between the RSU and *v* is built.

**Protocol 2** V2I authentication protocol.
1:*v*:   choose $PS_{i}$, $SK_{i}$, $EXP_{i}$;   compute         $V=r_{v}H_{1}(PS_{i}||EXP_{i})$;         $h=H_{5}(PS_{i}||EXP_{i}||TS_{1}\left|\right|N_{1}\left|\right|r_{v}P,V)$;         $W=\left(r_{v}+h\right)SK_{i}$;2:v→RSU: $PS_{i}$, $EXP_{i}$, $TS_{1}$, $N_{1}$, $r_{v}P$, and $sign_{v}=\left\{V,W\right\}$;3:RSU:   check $EXP_{i}$ and $TS_{1}$;   compute $h=H_{5}(PS_{i}||EXP_{i}||TS_{1}\left|\right|N_{1}\left|\right|r_{v}P,V)$ and $PK_{i}=H_{1}(PS_{i}||EXP_{i})$;   check $e\left(P,W\right)=e\left(P_{pub},V+hPK_{i}\right)$;   choose $r_{RSU}$∈$Z_{q}^{*}$;   compute         $V^{\prime }$=$r_{RSU}H_{1}(ID_{RSU}||EXP_{RSU})$;         $h^{\prime }$=$H_{5}(ID_{RSU}||EXP_{RSU}||TS_{2}\left|\right|N_{2}\left|\right|r_{RSU}P,V^{\prime })$;         $W^{\prime }$ = $\left(r_{RSU}+h^{\prime }\right)SK_{RSU}$;   set $K_{RSU-v}$=$r_{RSU}r_{v}P$;   encrypt $N_{1}$ to obtain $C_{RSU-v}$=$Enc\_K_{RSU-v}\left\{N_{1}\right\}$;4:RSU→v: $ID_{RSU}$, $EXP_{RSU}$, $TS_{2}$, $N_{2}$, $r_{RSU}P$, $sign_{RSU}$, $C_{RSU-v}$;5:*v*:   check $EXP_{RSU}$ and $TS_{2}$;   compute $h^{\prime }=H_{5}(ID_{RSU}||EXP_{RSU}||TS_{2}\left|\right|N_{2}\left|\right|r_{RSU}P,V^{\prime })$ and $PK_{RSU}=H_{1}(ID_{RSU}||EXP_{RSU})$;   check $e\left(P,W^{\prime }\right)=e\left(P_{pub},V^{\prime }+h^{\prime }PK_{RSU}\right)$;   set $K_{v-RSU}=r_{v}r_{RSU}P$;   encrypt $N_{2}$ to obtain $C_{v-RSU}$=$Enc\_K_{v-RSU}\left\{N_{2}\right\}$;6:v→RSU: $C_{v-RSU}$;7:RSU:   decrypt $C_{v-RSU}$;   verify $N_{2}$.


#### 4.6.2. Pseudonym Issuance Protocol

The pseudonym issuance protocol is presented as Figure 8 and Protocol 3. After finishing the V2I authentication, vehicle *v* is able to send the message to the RSU and apply for multiple temporary pseudonyms and certificates within the BS domain via a secure channel. When receiving the message from *v*, the RSU forwards the message to the BS. The BS is able to generate multiple new pseudonyms, public keys, private keys, certificates, and group keys for the vehicle. Afterwards, the BS computes the session key between the BS and vehicle, encrypts the message by the session key to generate the ciphertext, and sends the ciphertext, its identity $ID_{BS}$, and the public key $P_{pub}^{BS}$ to the vehicle via the RSU. When receiving the message from the BS, *v* computes the session key between *v* and the BS to decrypt the ciphertext to obtain multiple new pseudonyms, public keys, private keys, certificates, and group keys from the BS. Here, *v* is able to use the pseudonyms issued by the BS to communicate with other entities in the BS domain and change the pseudonyms regularly to improve its anonymity. The details are depicted as follows.
**Protocol 3** Pseudonym issuance protocol.1:*v*:compute $C_{v-RSU}=Enc\_K_{v-RSU}\left\{Req_{PS_{i}}\right\}$;2:v→RSU:   $C_{v-RSU}$, $PS_{i}$;3:RSU:   decrypt $C_{v-RSU}$ to obtain $Req_{PS_{i}}$;   compute $C_{RSU-BS}=Enc\_K_{RSU-BS}\{Req_{PS_{i}}||PS_{i}||EXP_{i}\left|\right|r_{v}P$};4:RSU→BS:   $ID_{RSU},C_{RSU-BS}$;5:BS:   decrypt $C_{RSU-BS}$ to obtain $Req_{PS_{i}}$, $EXP_{i}$, $PS_{i}$, and $r_{v}P$;   compute multiple $PS_{i}^{BS}$, $SK_{i}^{BS}$, $PK_{i}^{BS}$, $Cert_{i}^{BS}$ and $K_{BS}$, $K_{BS-v}$, $C_{BS-v}$:         $PS_{i}^{BS}\leftarrow {\left\{0,1\right\}}^{n}$;         $SK_{i}^{BS}=x_{i}\leftarrow Z_{q}^{*}$;         $PK_{i}^{BS}=x_{i}P$;         $Cert_{i}^{BS}$ =$\left\{\sigma_{1},\sigma_{i},w,s_{i}\right\}$;         where $\sigma_{1}=CC\_Sign\left\{w\right\}$;                     $\sigma_{i}=H\left(PS_{i}^{BS}\right)SK_{BS}^{\prime }+r(s_{i}H_{1}(ID_{BS}||EXP_{BS})+H\left(PS_{i}^{BS}\right)H_{2}\left(PK_{i}^{BS}\right))$;                      w=rP;                      $r,s_{i}\leftarrow Z_{q}^{*}$;         $K_{BS}\leftarrow {\left\{0,1\right\}}^{n}$;         $K_{BS-v}=r_{BS}r_{v}P$;         $C_{BS-v}=Enc\_K_{BS-v}\{PS_{i}^{BS}||SK_{i}^{BS}||PK_{i}^{BS}||Cert_{i}^{BS}\left|\right|K_{BS}\}$;   store $PS_{i}^{BS}$, $SK_{i}^{BS}$, $PS_{i}$, $EXP_{i}$ in pseudonym list;6:BS→v:   $ID_{BS}$, $C_{BS-v}$, $P_{pub}^{BS}$;7:*v*:   compute $K_{v-BS}=r_{v}P_{pub}^{BS}$ and decrypts $C_{BS-v}$;   store $PS_{i}^{BS}$, $PK_{i}^{BS}$, $SK_{i}^{BS}$, $Cert_{i}^{BS}$, $K_{BS}$, $ID_{BS}$, and $P_{pub}^{BS}$;


In order to apply for multiple temporary pseudonyms and certifications within the BS, vehicle *v* uses session key $K_{v-RSU}$ to encrypt request $Req_{PS_{i}}$ to obtain ciphertext $C_{v-RSU}=Enc\_K_{v-RSU}\left\{Req_{PS_{i}}\right\}$.Vehicle *v* sends $C_{v-RSU}$ to the RSU.When obtaining the ciphertext from vehicle *v*, the RSU uses session key $K_{RSU-v}$ to decrypt $C_{v-RSU}$ to obtain the request $Req_{PS_{i}}$. Then, the RSU uses the session key between the RSU and BS $K_{RSU-BS}$ to encrypt $Req_{PS_{i}}$, $PS_{i}$, $EXP_{i}$, and $r_{v}P$ and obtain $C_{RSU-BS}$.The RSU sends ciphertext $C_{RSU-BS}$ to the BS.The BS decrypts $C_{RSU-BS}$ and obtains $Req_{PS_{i}}$, $PS_{i}$, $EXP_{i}$, and $r_{v}P$. Then, multiple temporary pseudonyms $PS_{i}^{BS}\leftarrow {\left\{0,1\right\}}^{n}$, multiple random numbers $x_{i}\leftarrow Z_{q}^{*}$, group key $K_{BS}\leftarrow {\left\{0,1\right\}}^{n}$, and $r,s_{i}\leftarrow Z_{q}^{*}$ are selected and the private key $SK_{i}^{BS}$, public key $PK_{i}^{BS}$, and certificate $Cert_{i}^{BS}$ are computed, where
$$\begin{array}{cc} SK_{i}^{BS} & =x_{i},\\ PK_{i}^{BS} & =x_{i}P,\\ w & =rp,\\ \sigma_{1} & =CC\_sign\{w\},\\ \sigma_{i} & =H\left(PS_{i}^{BS}\right)SK_{BS}^{\prime }+r(s_{i}H_{1}(ID_{BS}||EXP_{BS})+H\left(PS_{i}^{BS}\right)H_{2}\left(PK_{i}^{BS}\right)),\\ Cert_{i}^{BS} & =\{\sigma_{1},\sigma_{2},w,s_{i}\} \end{array}$$The BS sets the session key $K_{BS-v}=r_{BS}r_{v}P$ and encrypts $PS_{i}^{BS}$, $SK_{i}^{BS}$, $PK_{i}^{BS}$, and $K_{BS}$ to obtain ciphertext $C_{BS-v}$=$Enc\_K_{BS-v}\{PS_{i}^{BS}||SK_{i}^{BS}||PK_{i}^{BS}\left|\right|K_{BS}\}$. Finally, the BS stores $PS_{i}^{BS}$, $SK_{i}^{BS}$, $PS_{i}$, $EXP_{i}^{BS}$.The BS sends $C_{BS-v}$, $ID_{BS}$, and $P_{pub}^{BS}$ to vehicle *v* via the RSU.After receiving the ciphertext from the BS, vehicle *v* computes the session key $K_{v-BS}=r_{v}P_{pub}^{BS}$ and decrypts $C_{BS-v}$ to obtain the message from the BS. Finally, vehicle *v* stores $PS_{i}^{BS}$, $PK_{i}^{BS}$, $SK_{i}^{BS}$, $Cert_{i}^{BS}$, $K_{BS}$, $ID_{BS}$, and $P_{pub}^{BS}$ locally.


### 4.7. Pseudonym Change Protocol

When vehicle $v_{i}$ runs on the road, it is requested to broadcast the BSM with $PS_{i}^{BS}$. If meeting other vehicles in the BS domain, vehicle $v_{i}$ believes that there is a chance to change its pseudonym. Now, $v_{i}$ is able to broadcast a pseudonym change request and try to communicate with other vehicles in the vicinity to change its pseudonym. Different from the traditional mix zone mechanism, the proposed pseudonym change protocol does not need the assistance of the RSUs, which means all vehicles in the BS domain can change their pseudonyms independently.

The pseudonym change protocol includes two periods: pseudonym sharing period and pseudonym change period. In the pseudonym sharing period, vehicles share their own stored pseudonyms, certificates, and driving status. If the number of pseudonyms received is not enough or there are considerable differences in the driving among vehicles, vehicles only store the information received. Otherwise, vehicles store the information received and start the pseudonym change period. In this period, all vehicles change their pseudonyms and communicate with other entities as group members. The details of the pseudonym change protocol are depicted as Figure 9 and Protocol 4:Vehicle $v_{i}$ selects pseudonym $PS_{i}^{BS}$, public key $PK_{i}^{BS}$, and certificate $Cert_{i}^{BS}$ and computes signature $Sign_{v_{i}}=sign\_SK_{i}^{BS}\{PS_{i}^{BS}||PK_{i}^{BS}||TS_{i}\left|\right|t_{start}\left|\right|t_{end}\left|\right|t_{change}\left||change\_request\right\}$, where $TS_{i}$ is the current timestamp and change_request is the pseudonym change request.Vehicle $v_{i}$ broadcasts $PS_{i}^{BS}$, $PK_{i}^{BS}$, $TS_{i}$, $t_{start}$, $t_{end}$, $t_{change}$, $Cert_{i}^{BS}$, $Sign_{v_{i}}$, and change_request to other surrounding vehicles.When receiving the request from vehicle $v_{i}$, the vehicle in the vicinity (e.g., $v_{j}$) checks the freshness of timestamp $TS_{i}$ and the legality of signature $Sign_{v_{i}}$. If the above verification is successful, $v_{j}$ updates the pseudonym certificate list $PS-Cert-List_{j}$ and computes the ciphertext $Msg=Enc\_K_{BS}\left\{PS-Cert-List_{j}\right\}$.When the current time $t\geq t_{start}$ and $t<t_{end}$, $v_{j}$ broadcast the ciphertext to surrounding vehicles.Surrounding vehicles $v_{k}$ (including $v_{i}$) decrypt Msg and add $PS_{j}^{BS}$, $PK_{j}^{BS}$, and $Cert_{j}^{BS}$ into $PS-Cert-List_{k}$.Finally, if num≥threshold, all vehicles compute $Cert^{BS}=\sum_{i=1}^{num}Cert_{i}^{BS}$=$\left\{\sigma_{1},\sigma_{2}\right\}$, where $\sigma_{2}=\sum_{i=1}^{num}\sigma_{i}$, $PK^{BS}=PK^{BS}\cup PK_{j}^{BS},PS^{BS}=PS^{BS}\cup PS_{i}^{BS}$, and change pseudonym $PS^{BS}$ and certificate $Cert^{BS}$ after $t_{change}$.
**Protocol 4** Pseudonym change protocol.1:$v_{i}$:compute $Sign\_SK_{i}^{BS}\{PS_{i}^{BS}||PK_{i}^{BS}||TS_{i}\left|\right|t_{start}\left|\right|t_{end}\left|\right|t_{change}\left||change\_request\right\}$2:$v_{i}\to$ other vehicle (e.g., $v_{j}$):      broadcast $PS_{i}^{BS}$, $PK_{i}^{BS}$, $TS_{i}$, $t_{start}$, $t_{end}$, $t_{change}$, $Cert_{i}^{BS}$, $Sign_{v_{i}}$, change_request;3:$v_{j}$:      verify pseudonym changing request $request_{i}$;      update and add new $PS_{j}^{BS}$, $PK_{j}^{BS}$, $Cert_{j}^{BS}$ into $PS-Cert-List_{j}$;      compute $Msg=Enc\_K_{BS}\left\{PS-Cert-List_{j}\right\}$;4:$v_{j}\to$ surrounding vehicle (e.g., $v_{k}$): Msg5:$v_{k}$:      decrypt Msg;      add $PS_{j}^{BS}$, $Cert_{j}^{BS}$ from $PS-Cert-List_{j}$ into $PS-Cert-List_{k}$;      compute            $Cert^{BS}=\sum_{i=1}^{num}Cert_{i}^{BS}$=$\left\{\sigma_{1},\sigma_{2}\right\}$, where $\sigma_{2}=\sum_{i=1}^{num}\sigma_{i}$;            $PS^{BS}=PS^{BS}\cup PS_{i}^{BS}$;            $PK^{BS}=PK^{BS}\cup PK_{j}^{BS}$;6:all vehicles:**if** 
num≥threshold 
**then**      all vehicles change pseudonym $PS^{BS}$ and certificate $Cert^{BS}$ after $t_{change}$;**end if**

The number of vehicles changing pseudonyms depends on the current user-centric location privacy level (as depicted in Section 5.3.2). When the location privacy level of the vehicle is low, the vehicle has to share more pseudonyms. When the user-centric location privacy is at a high level, the vehicle does not need to sacrifice too many pseudonyms to protect its privacy. In addition, due to the limited communication range of the BS, when the vehicle is driven from one BS (e.g., $BS_{1}$) to another BS (e.g., $BS_{2}$), the vehicle is required to reapply to $BS_{2}$ for the new pseudonym list. Therefore, the number of pseudonyms only needs to guarantee the privacy and security of vehicles within the BS domain.

### 4.8. Pseudonym Revocation Protocol

Generally, the pseudonym revocation protocol is used in the following conditions: (1) The vehicle’s pseudonym and certificate have expired. In the pseudonym issuance protocol, $K_{BS}$ is required to be regularly updated by the BS and the period of the availability of $K_{BS}$ cannot be longer than that of $EXP_{i}$. Since the BS issues enough pseudonyms to the vehicles, the validity period of $K_{BS}$ can be set long enough, which can reduce the communication overhead caused by the frequent requests for new pseudonyms. However, once $K_{BS}$ or $EXP_{i}$ expires, vehicles have to reapply for new pseudonyms from the BS or TA. (2) Legal vehicles are compromised. In PCP, two cuckoo filters [32] are used and maintained by the BS: the positive filter posfilter and the negative filter negfilter, where posfilter stores the valid pseudonyms and negfilter stores the illegal pseudonyms. After receiving the illegal vehicles’ information including signature σ, message *M*, multiple group pseudonyms $PS^{BS}$, and proof, the BS queries the local pseudonym list and obtains the multiple private keys $SK^{BS}$ according to $PS^{BS}$ firstly. Then, the BS computes the signatures of *M* to obtain $\{\sigma_{1}$, $\sigma_{2}$, ..., $\sigma_{n}\}$, respectively. If $\sigma_{i}=\sigma$, the BS believes that the pseudonym $PS_{i}^{BS}$ and private key $SK_{i}^{BS}$ corresponding to $\sigma_{i}$ are the identity information of the illegal vehicle. After that, the BS selects all pseudonyms $PS_{i}$ issued by the BS and the pseudonym issued by the TA about the illegal vehicle, removes these pseudonyms from posFilter, and adds them to negFilter to exclude the illegal vehicles from the VANET. The BS further broadcasts two filters in the BS domain via the RSU. Finally, the BS sends $PS_{i}$ to the TA and reveals the real identity of the illegal vehicle. When receiving the message from the BS, the TA decrypts $PS_{i}$ to obtain the real identity $ID_{i}$. The TA sends all pseudonyms related to the illegal vehicles to the BS and prevents the illegal vehicles from reapplying for new pseudonyms.

## 5. Performance Analysis

In this section, we discuss the performance of the proposed scheme in V2I authentication in terms of the computational cost and communication cost compared with LIAP [15] and SPA [25]. In addition, the Veins simulation framework is adopted to conduct the simulation experiment in terms of the average anonymous set size and user-centric location privacy level to manifest the security of the proposed pseudonym change protocol.

### 5.1. Computation Cost

The computational cost is defined to evaluate the total computation time required for pseudonym change, which is mainly dominated by hash-to-point ($T_{mtp}$), point exponentiation ($T_{pe}$), point multiplication ($T_{pm}$), and bilinear pairing ($T_{bp}$) all over the group.

In LIAP, given system parameters {P, *q*, $G_{1}$, $G_{2}$, *e*, $PK_{CA}$, *H*, h}, where $H:{\left\{0,1\right\}}^{*}\to Z_{q}^{*}$ and *h* is the one-way hash function, such as SHA-2, the RSU broadcasts message $M=\{PK_{R}$, $Cert_{R}$, *T*, $RPK_{i}^{1}$, $RPK_{i}^{2}$, $RPK_{i-1}^{1}$, $RPK_{i-1}^{2}$, $RPK_{i+1}^{1}$, $RPK_{i+1}^{2}\}$, and $\sigma_{r}$, where $Cert_{R}=Sign_{CA}\left\{PK_{R}\right\}$ is the certificate of the public key $PK_{R}$ of the RSU, *T* is the timestamp of the message, $RPK_{i}^{1},RPK_{i}^{2},RPK_{i-1}^{1},RPK_{i-1}^{2},RPK_{i+1}^{1},RPK_{i+1}^{2}$ are the RSU local public keys, and $\sigma_{r}$ is the signature of message *M*. When receiving the message from the RSU, vehicle *v* is required to verify the legality of $Cert_{R}$ and $\sigma_{r}$. If the message from the RSU is legal, *v* uses $PK_{R}$ to encrypt the vehicle’s public key $PK_{v}$, certificate $Cert_{v}$, timestamp $T^{\prime }$, and signature $\sigma_{v}=Sign_{v}\left\{PK_{v},Cert_{v},T\right\}$ and obtains $C_{v-RSU}$. When receiving the message from the vehicle, the RSU first decrypts $C_{v-RSU}$ and checks $T^{\prime }$. Then, the RSU verifies the legality of $Cert_{v}$ and $\sigma_{v}$. If the above verification is successful, the RSU believes that *v* is legal; otherwise, the message will be dropped. Since LIAP does not give the detail of the certification and signature generation mechanisms, we adopted the same CC signature mechanism as PCP and the BF-IBE encryption algorithm to derive the computational cost of LIAP.

In SPA, the RSU is used as the fog–edge node (FEN) to provide the communication service for the vehicles. Given public parameter $\left\{p,q,a,b,G_{1},G_{2},e,P,Q,Q^{\prime },h_{1},h_{2},h_{3}\right\}$, when entering in the communication range of the RSU, vehicle *v* is able to send $PID_{v}$, $M_{v}$, and signature $\mu_{v}$ to the RSU for authentication, where $\mu_{v}=\left\{T_{v},U_{v}\right\}$, $T_{v}=S_{v}H_{i}+k_{i}Q^{\prime }$, $U_{v}=k_{i}P$, $S_{v}$ is the private key of the vehicle, $H_{i}=h_{2}(PID_{v}\left|\right|M_{v})$, $k_{i}\leftarrow Z_{q}^{*}$, and timestamp TS is in $M_{v}$. When receiving the message $PID_{v}$, $M_{v}$, and $\mu_{v}$, the RSU first computes $H_{j}=h\left(RID_{FEN}\right)$ and $H_{i}=h_{2}(PID_{v}\left|\right|M_{v})$. Then, the RSU verifies whether the equation $e\left(T_{v},P\right)=e\left(P_{Pub}H_{j}H_{i},Q\right)e\left(U_{v},Q^{\prime }\right)$ holds. If it does not hold, the RSU discards the message; otherwise, the RSU believes that *v* is legal. Then, the RSU sends $RID_{FEN}$, $M_{FEN}$ and signature $\mu_{FEN}=\left\{T_{FEN},U_{FEN}\right\}$ to *v*, where timestamp TS is in $M_{FEN}$, $T_{FEN}=S_{FEN}H+kQ^{\prime }$, $U_{FEN}=k_{i}P$, $S_{FEN}$ is the private key of the RSU, and $H_{i}=h_{2}(RID_{FEN}\left|\right|M_{v})$, $k\in Z_{q}^{*}$. *v* firstly checks the freshness of TS and calculates $H_{j}=h_{1}\left(RID_{FEN}\right)$ and $H_{i}=h_{2}(RID_{FEN}\left|\right|M_{v})$. Then, *v* verifies whether $e\left(T_{FEN},P\right)\overset{?}{=}e\left(Pub_{FEN}H_{j}H_{i},Q\right)e(U_{FEN},$$Q^{\prime })$ holds. If it does hold, *v* believes that the RSU is legal; otherwise, the message from the RSU is discarded.

In PCP, vehicle *v* generates signature $sign_{v}=\left\{V,W\right\}$ and sends $PS_{i}$, $EXP_{i}$, $TS_{1}$, $N_{1}$, $r_{v}P$, $sign_{v}$ to the RSU, where $V=r_{v}H_{1}(PS_{i}||EXP_{i})$, $h=H_{5}(PS_{i}||TS_{1}\left|\right|N_{1}\left|\right|r_{v}P,V)$, $W=\left(r_{v}+h\right)SK_{i}$. When obtaining the message from *v*, the RSU computes *h* and $PK_{i}$. Then, the RSU checks the equation $e\left(P,W\right)\overset{?}{=}e\left(P_{pub},V+hPK_{i}\right)$. If the equation holds, the RSU believes *v* is legal. Afterwards, the RSU signs $ID_{RSU}$, $EXP_{RSU}$, $TS_{2}$, $N_{2}$, and $r_{RSU}P$ to obtain $sign_{RSU}=\left\{V^{\prime },W^{\prime }\right\}$, where $V^{\prime }$=$r_{RSU}H_{1}(ID_{RSU}||EXP_{RSU})$, $h^{\prime }$=$H_{5}(ID_{RSU}||TS_{2}\left|\right|N_{2}||r_{RSU}P,V^{\prime })$, $W^{\prime }$=$\left(r_{RSU}+h^{\prime }\right)SK_{RSU}$. After that, the RSU sends $ID_{RSU}$, $TS_{2}$, $N_{2}$, $r_{RSU}P$, $sign_{RSU}$, and $C_{RSU-v}$ to *v*. *v* computes $h^{\prime }$, $PK_{RSU}$ and verifies the legality of the RSU by checking $e\left(P,W^{\prime }\right)\overset{?}{=}e\left(P_{pub},V^{\prime }+h^{\prime }PK_{RSU}\right)$.

Table 2 depicts the comparisons of the computational costs for the vehicle and RSU. In LIAP, in order to protect the public key and certificate of the vehicle from being exposed, the vehicle is requested to use the public key of the RSU to encrypt its certificate and public key, which leads to extra computational cost. Since the complicated signature mechanism is adopted, the V2I authentication protocol in SPA requires the vehicle and RSU to execute more point multiplications and bilinear pairing operations, causing a high computational cost. In PCP, the identity-based signature mechanism is adopted, so the computational cost of the hash-to-point operation becomes the vital factor for the efficiency of V2I authentication.

### 5.2. Communication Cost

The communication cost refers to the total size of the message transmitted during authentication. According to [33], the size of each single element in $G_{1}$ and $Z_{q}^{*}$ is 128 bytes and 40 bytes, respectively. The sizes of the expiration and timestamp are 4 bytes. In the authentication protocol, since LIAP and SPA only transmit authentication-related messages and ignore the necessary messages to establish a secure channel, we only considered the communication cost related to V2I authentication.

In LIPA, the RSU broadcasts message $M=\{PK_{R}$, $Cert_{R}$, *T*, $M_{RPK}\}$, $\sigma_{r}$, where $|PK_{R}\left|=\right|G_{1}|$, $|Cert_{R}\left|=2\right|G_{1}|$. The vehicle sends $C_{v-RSU}$ to the RSU, where $|C_{v-RSU}\left|=3\right|G_{1}\left|+|TS\right|$. Thus, the total communication cost of LIAP is:$$\begin{array}{c} 6\times |G_{1}\left|+2|TS\right| \end{array}$$

In SPA, the vehicle needs to send $PID_{v}$, $M_{v}$, and signature $\mu_{v}=\left\{T_{v},U_{v}\right\}$ to the RSU for authentication, where $|PID_{v}\left|=\right|Z_{q}^{*}|$, $|T_{v}\left|=\right|U_{v}\left|=\right|G_{1}|$. After verifying the message, the RSU sends $RID_{FEN}$, $M_{FEN}$ and signature $\mu_{FEN}=\{T_{FEN}$, $U_{FEN}\}$ to the vehicle, where $|RID_{FEN}\left|=\right|Z_{q}^{*}|$, $|T_{FEN}\left|=\right|U_{FEN}\left|=\right|G_{1}|$. Consequently, the communication cost of SPA is:$$\begin{array}{c} 4\times |G_{1}\left|+2|TS|+2\times \right|Z_{q}^{*}| \end{array}$$

In PCP, the vehicle sends $PS_{i}$, $TS_{1}$, and $sign_{v}$ to the RSU. When finishing the verification of the received message, the RSU sends $ID_{RSU}$, $TS_{2}$, and $sign_{RSU}$ to the vehicle, where $|PS_{i}|=|ID_{RSU}|$ = $|Z_{q}^{*}|$, $|sign_{v}|=|sign_{RSU}\left|=2\right|G_{1}|$. Thus, the communication cost of PCP is:$$\begin{array}{c} 4\times |G_{1}\left|+2|TS|+2\times \right|Z_{q}^{*}| \end{array}$$

We can see that PCP and SPA have a low communication cost. In LIPA, the vehicle and RSU are requested to send extra certificates and public keys, which causes a high communication cost.

### 5.3. Simulation

In this section, Veins [34] is introduced to evaluate PCP in terms of the average anonymous set size and the average strength of location privacy. The proposed protocols were implemented using C++, where the experimental environment included a 2.6 GHz Intel(R) Core(TM) i7-6700HQ CPU, 2GB RAM, and the Debian 9.4 operating system. The Pairing Based Cryptography Library [35] was adopted to implement the cryptographic operations. We used the Veins simulation framework to conduct extensive simulations, through the tools of SUMO and OMNET++. A SUMO network file was edited to simulate the scenario of pseudonym change depicted in Figure 5 (Simulation 1). In addition, a road map of Xi’an from OpenStreetMap (OSM) [36] was chosen as the real simulation scenario (Simulation 2). The road map from OSM can be converted into the network file by NETCONVERT. POLYCONVERT was used to generate the topographic file. The RandomTrips Python script was adopted to generate random vehicle trips. A SUMO configuration file was edited to integrate the network file, topographic file, and vehicle trips file. The SUMO simulation from the real map and Veins simulation are shown in Figure 10. The parameters used in the simulations are shown in Table 3.

#### 5.3.1. Average Anonymous Set Size

The average anonymous set size is defined as the set of available candidate pseudonyms that are used in the pseudonym change protocol [14]. The larger the set of pseudonyms, the better it is able to confuse the tracking of GPAs. Figure 11 and Figure 12 show the average anonymous set size in Simulation 1 and Simulation 2, respectively.

In Figure 11, the vehicle switches to the different scenarios shown in Figure 5 every 15 s. In Scenario 1 of Figure 5, since there are not enough vehicles to guarantee that the vehicles’ anonymity set meets the pseudonym change security requirements and there is no RSU to provide pseudonym change support, the PCP, mix zone and silent period schemes cannot support pseudonym change in Scenario 1. However, in order to evaluate Scenario 1 for the effect of pseudonym change in subsequent scenarios, Scenario 1 is supplemented during the switching process of Scenarios 2–4. In Scenario 2, the above three schemes are able to support pseudonym change. Mix zones support more vehicles participating in pseudonym change than silent periods due to the wider communication range of RSUs. Since the vehicle collects more pseudonyms form other vehicles in Scenario 1, PCP guarantees providing larger pseudonym sets in Scenario 2. In Scenario 3, due to the low numbers of vehicles, the silent period scheme cannot protect the location privacy of vehicles that want to change pseudonyms. Because the RSU is not deployed, mix zones cannot provide the pseudonym change service for vehicles in Scenario 4.

In Figure 12, we can see that the average anonymous set size increases rapidly due to the pseudonym change protocol. However, more notably, we observe that the vehicle density and traffic conditions have a significant impact on the anonymous set size since the vehicles have a greater chance to communicate with surrounding vehicles and share local pseudonym sets. The denser traffic conditions make it easier for the vehicle to meet other vehicles with a similar driving status. Moreover, the average anonymous set size in mix zones depends entirely on the deployment density of the RSUs. However, the high-density deployment of the RSUs requires a very high cost in a short time, which makes the silent period and PCP more suitable for the actual traffic scene. Meanwhile, as it depends on the number of pseudonyms shared by the vehicles in the communication group rather than the number of vehicles, PCP has a higher average anonymous set size compared with the silent period.

#### 5.3.2. User-Centric Location Privacy Level

The user-centric location privacy level [37] of the vehicles in VANETs is modeled by the location privacy loss function $\beta_{i}\left(t,T_{i}\right):\left(IR^{+},IR^{+}\right)\to IR^{+}$, where *t* and $T_{i}$ refer to the current time and the time when $v_{i}$ changes pseudonym successful. According to a sensitivity parameter $0<\lambda_{i}<1$, the privacy loss is set 0 initially and increases with the time. The higher the value of $\lambda_{i}$, the faster the rate of privacy loss is. The privacy loss function is defined as
$$\begin{array}{c} \beta_{i}\left(t,T_{i}\right)=\left\{\begin{array}{cc} \lambda_{i}\cdot \left(t-T_{i}\right) & \mathrm{for}\;T_{i}\leq t<T_{i}^{max}\\ A_{i}\left(T_{i}\right) & \mathrm{for}\;T_{i}^{max}\leq t \end{array}\right. \end{array}$$
where $T_{i}^{max}$=$\frac{A_{i}\left(T_{i}\right)}{\lambda_{i}}+T_{i}$ refers to the time when the privacy loss function arrives at the maximal value. Given the location privacy loss function, the user-centric location privacy level of vehicle $v_{i}$ at time *t* is
$$\begin{array}{c} A_{i}\left(t\right)=A_{i}\left(T_{i}\right)-\beta_{i}\left(t,T_{i}\right),t\geq T_{i} \end{array}$$
Since vehicles cannot compute $A_{i}\left(T_{i}\right)$, an approximation $log_{2}\left(n\right)$ was used in the simulation [37].

Figure 13 gives the result of the user-centric location privacy level in different scenarios from Figure 5, where λ is defined as 0.1 and 0.8. Before changing the pseudonym, the user-centric location privacy level of each vehicle decreases linearly. Consequently, in Scenario 1 and other scenarios that do not meet the pseudonym change, the user-centric location privacy level of each vehicle gradually decreases and rises after changing pseudonym. Moreover, since the user-centric location privacy level is positively correlated with anonymous integration, the growth trend of the location privacy protection level is consistent with Figure 11.

Figure 14 shows the changes in the user-centric location privacy level of PCP, mix zones, and silent periods under different traffic conditions, respectively. We can see that the location privacy level increases dramatically at the beginning and remains stable after about 40 s. The greater the number of vehicles, the shorter the time for the user-centric location privacy level to reach the high level. Since silent periods and mix zones have more stringent requirements on pseudonym change conditions (e.g., slow speed, RSU deployment), PCP is able to improve the location privacy level faster than the other two schemes and maintains a high location privacy level.

## 6. Discussion

Aiming at the uneven distribution of vehicle density and low-density RSU deployment, this paper proposed a pseudonym change scheme for location privacy preserving in VANETs (PCP). PCP follows the 1609 standard proposed by the IEEE and is able to effectively guarantee the protection of the privacy of vehicles. However, there are several open problems that need to be addressed to support the large-scale deployment of VANETs:**Mac address change:** PCP supports the pseudonym change in the application layer. However, according to the 1609.4 standard [38], in order to protect the full location privacy and security of the vehicle, it is necessary to propose an effective mechanism to support the change of MAC address. Otherwise, only the pseudonym is changed, and the adversaries can still be associated with the tracked vehicle through the MAC address.**Beacon interval:** According to DSRC, each vehicle periodically broadcasts a BSM every 100–300 milliseconds [39,40]. Thus, the period of pseudonym change has to be limited to the beacon interval. However, a long time interval may cause the vehicle to be unable to obtain the driving status of the surrounding vehicles in time, and a short time interval cannot guarantee that the vehicle has enough time to change its pseudonym through cooperation. It is vital for VANETs to support an efficient beacon strategy.**Non-cooperative behavior:** The cooperation among vehicles is a key factor for a successful pseudonym changing strategy. However, due to the costs that are involved in changing the pseudonym, some vehicles may not be willing to cooperate with other vehicles. Therefore, how to improve the willingness of vehicles to change pseudonyms and ensure that the pseudonyms can be changed at a high location privacy level need to be further researched.

## 7. Conclusions

In this paper, we proposed a pseudonym change scheme for location privacy preserving in VANETs (PCP) to address the issue of location privacy. PCP first proposes a registration protocol, authentication protocol, and pseudonym revocation protocol to guarantee that all legal vehicles are able to communicate with surrounding entities and compromised vehicles can be revealed in time. Furthermore, we improved the ID-based linearly homomorphic signature scheme to support vehicle pseudonym change in various conditions, which can protect vehicle location privacy more effectively. Security and performance analysis showed that PCP is able to resist attacks from GPAs and keep a high location privacy level.

Our work leaves several open problems to be solved, for example designing an efficient signature mechanism to support vehicle anonymous communication and how many pseudonyms should a vehicle store to keep the balance between vehicle communication security and performance. In the future, we will focus on the above issues.

## Figures and Tables

**Figure 1 entropy-24-00648-f001:**
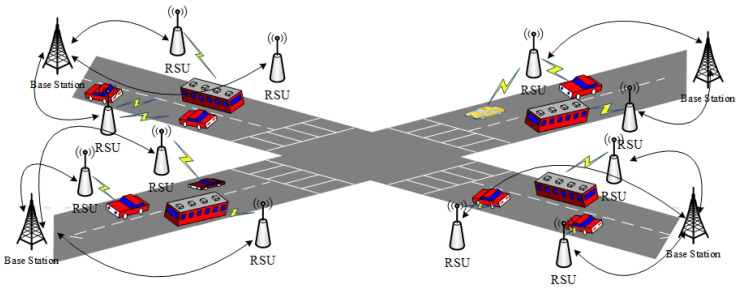
The framework of VANETs.

**Figure 2 entropy-24-00648-f002:**
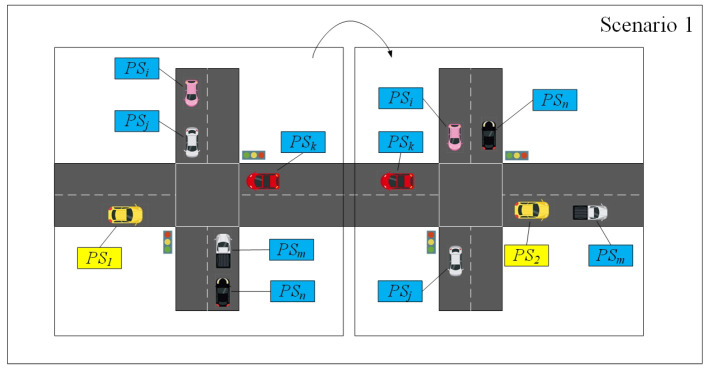
The syntactic linking scenario.

**Figure 3 entropy-24-00648-f003:**
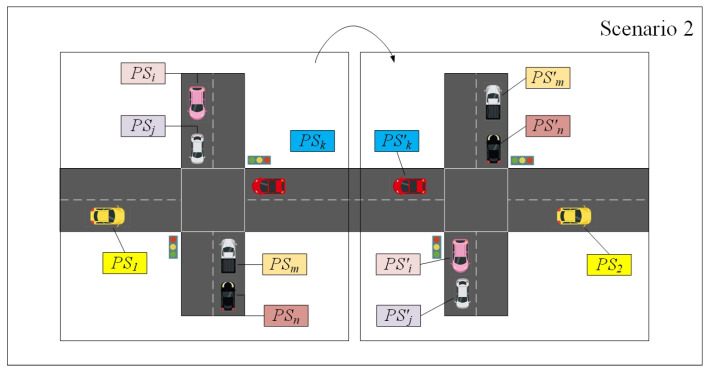
The semantic linking scenario.

**Figure 4 entropy-24-00648-f004:**
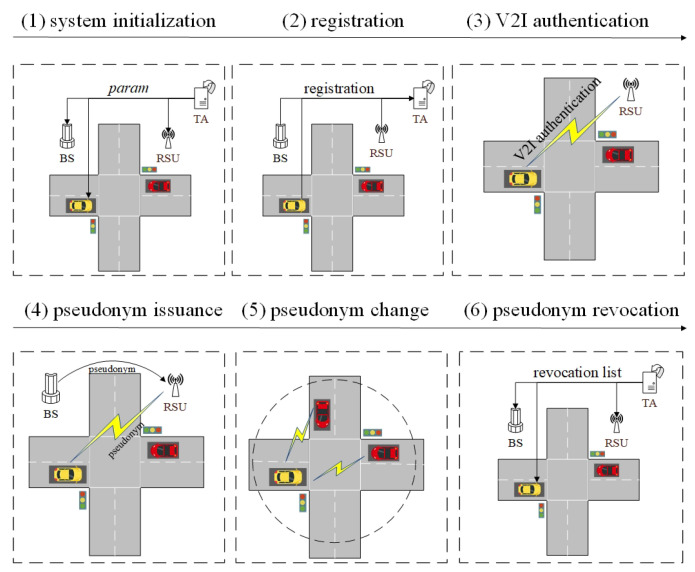
The scenario and participating entities of each protocol.

**Figure 5 entropy-24-00648-f005:**
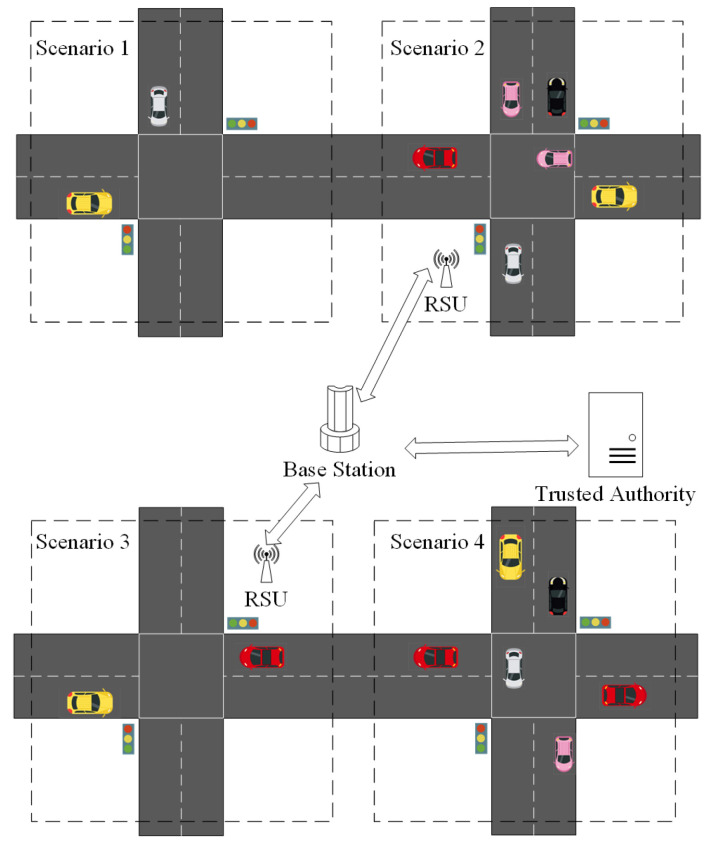
Scenarios of pseudonym change.

**Figure 6 entropy-24-00648-f006:**
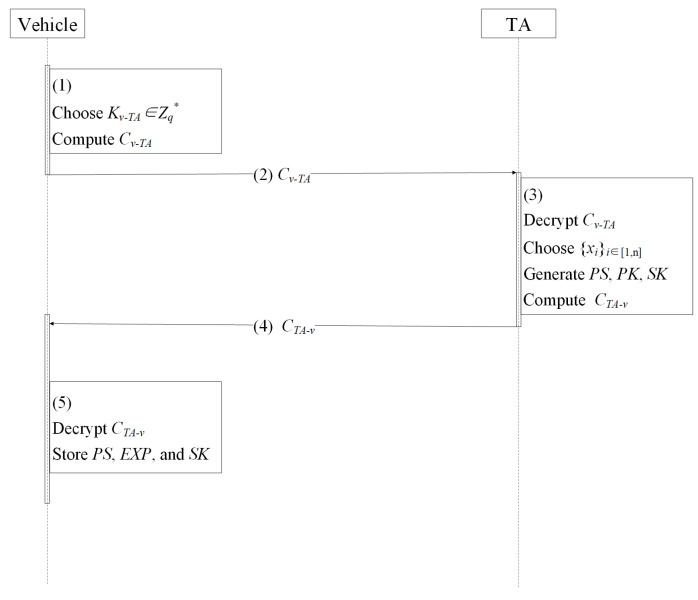
Vehicle registration protocol.

**Figure 7 entropy-24-00648-f007:**
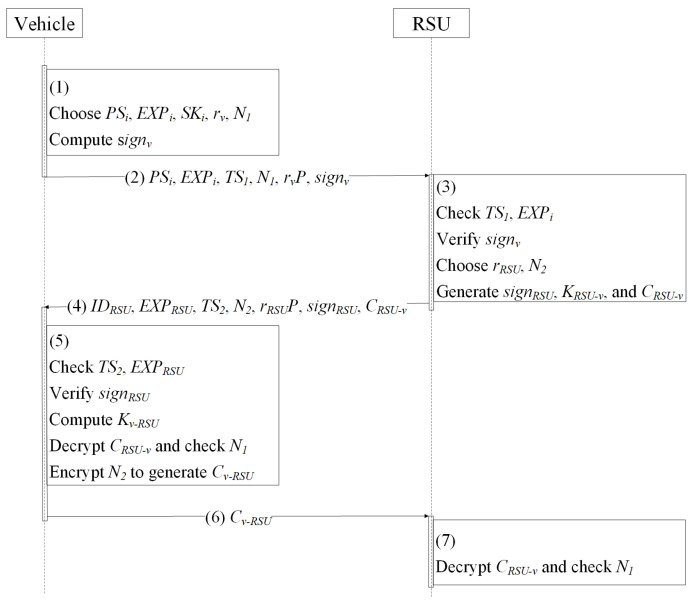
V2I authentication protocol.

**Figure 8 entropy-24-00648-f008:**
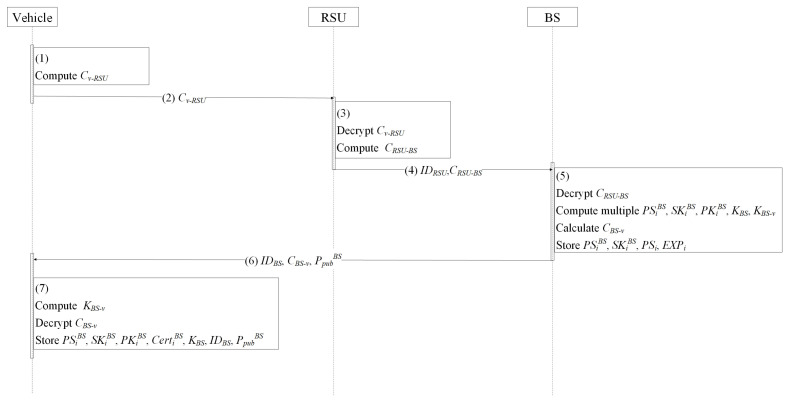
Pseudonym issuance protocol.

**Figure 9 entropy-24-00648-f009:**
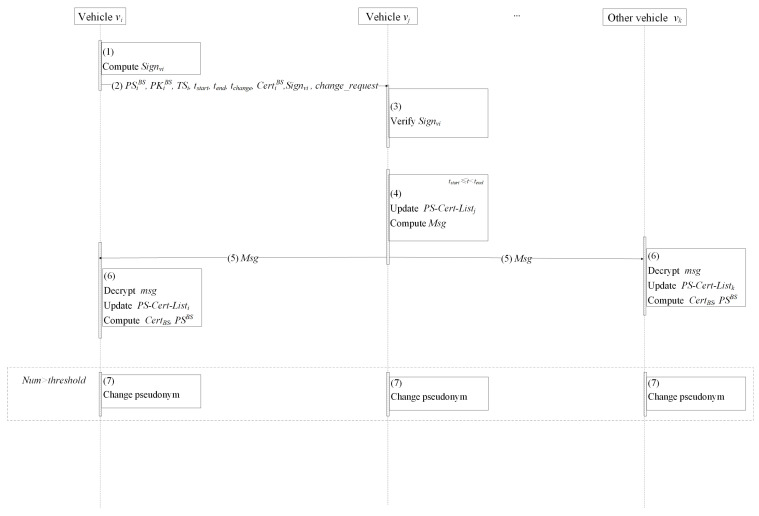
Pseudonym change protocol.

**Figure 10 entropy-24-00648-f010:**
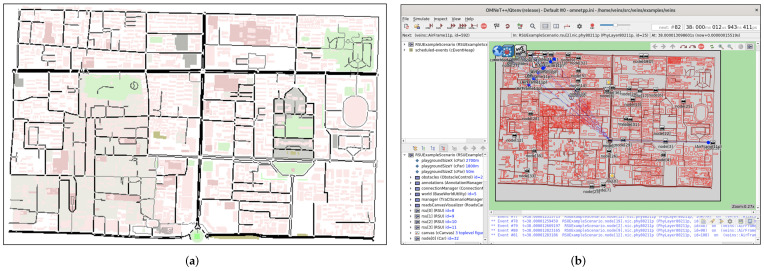
SUMO simulation and Veins simulation (Simulation 2). (**a**) SUMO simulation; (**b**) Veins simulation.

**Figure 11 entropy-24-00648-f011:**
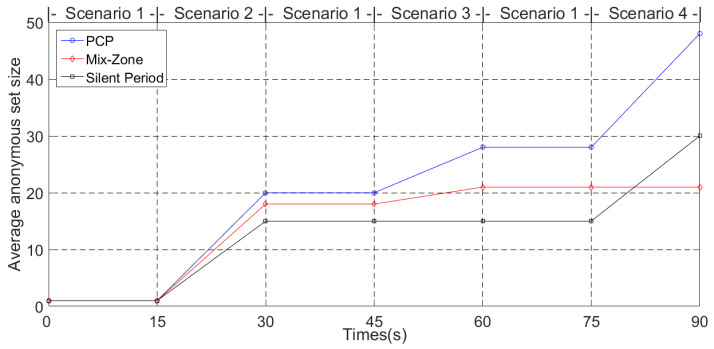
Average anonymous set size (Simulation 1).

**Figure 12 entropy-24-00648-f012:**
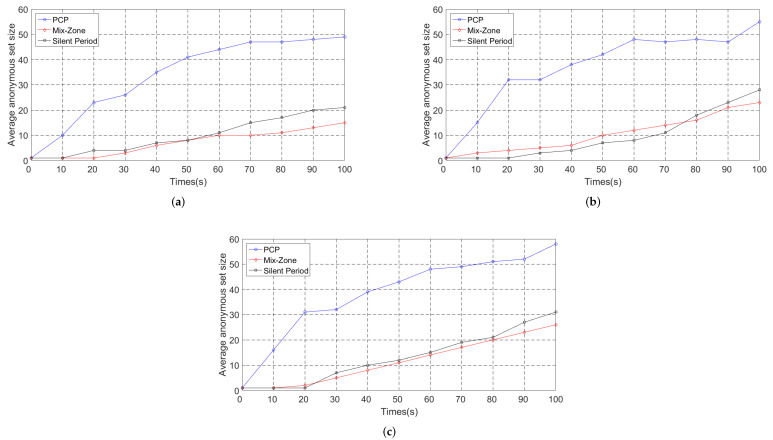
Average anonymous set size (Simulation 2). (**a**) Vehicle number *N* = 50. (**b**) Vehicle number *N* = 100. (**c**) Vehicle number *N* = 300.

**Figure 13 entropy-24-00648-f013:**
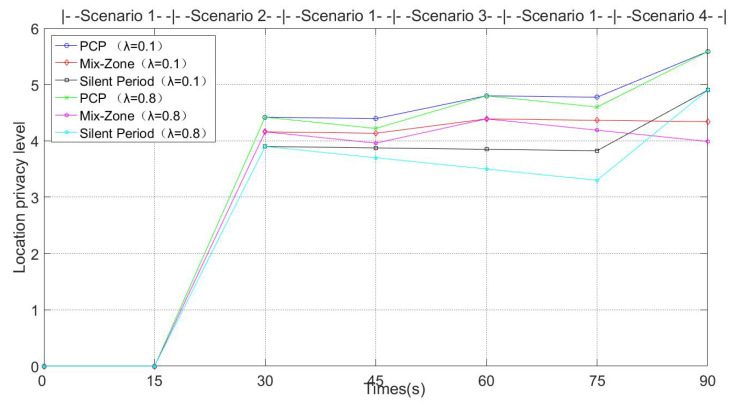
User-centric location privacy level (Simulation 1).

**Figure 14 entropy-24-00648-f014:**
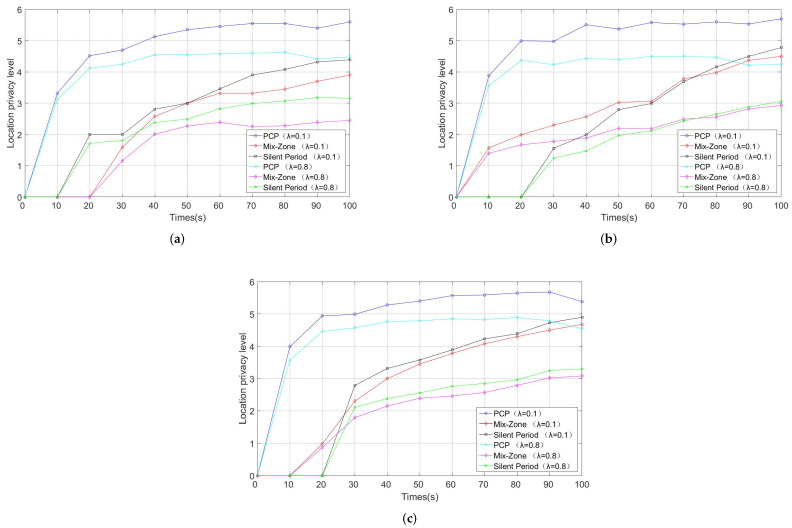
User-centric location privacy level (Simulation 2). (**a**) Vehicle number *N* = 50. (**b**) Vehicle number *N* = 100. (**c**) Vehicle number *N* = 300.

**Table 1 entropy-24-00648-t001:** The notations and descriptions used in this paper.

Notation	Description
$ID_{A}$	The real identity of entity A.
$PS_{i}$	The *i*th pseudonym of the vehicle issued by the TA. Each vehicle owns *n* pseudonyms $PS={\left\{PS_{i}\right\}}_{i\in n}$.
$PK_{i}/SK_{i}$	The public and private key pair of vehicle A’s pseudonym $PS_{i}$.
$PS_{i}^{BS}$	The *i*th pseudonym of the vehicle issued by the base station. Each vehicle owns *w* pseudonyms $PS^{BS}={\left\{PS_{i}^{BS}\right\}}_{i\in w}$.
$K_{A-B}$	The session key between entity A and entity B.
$Cert_{PS_{i}}^{BS}$	The *i*th certification of $PS_{i}^{BS}$ generated by the base station.
$TS_{i}$	The *i*th current timestamp.
$N_{i}$	The *i*th challenge value.
EXP	The expiration of the pseudonym.
$H_{i}$	The *i*th hash function.
$Sign\_SK_{i}\left\{M\right\}$	Sign message *M* with the private key $SK_{i}$.
$Sign_{A}$	The signature generated by entity A.
Enc_K{M}	Encrypt message *M* with the key *K*.
$C_{A-B}$	The ciphertext generated by entity A and the ciphertext sent to entity B.
num	The number of responses received by the vehicle when it sends a pseudonym change request.
$t_{start}$, $t_{end}$, $t_{change}$	The start time, the end time of pseudonym broadcast, and the pseudonym change time, respectively.

**Table 2 entropy-24-00648-t002:** Comparison of the computational cost.

Algorithm	LIAP	SPA	PCP
RSU-Gen	2$T_{PM}$	3$T_{PM}$	3$T_{PM}$
RSU-Ver	5$T_{BP}$ + 3$T_{PM}$	3$T_{BP}$ + 3$T_{PM}$	2$T_{BP}$ + $T_{PM}$ + $T_{MTP}$
V-Gen	$T_{BP}$ + 3$T_{PM}$ + $T_{PE}$	3$T_{PM}$	3$T_{PM}$
V-Ver	4$T_{BP}$ + 2$T_{PM}$	3$T_{BP}$ + 3$T_{PM}$	2$T_{BP}$ + $T_{PM}$ + $T_{MTP}$
Total	10$T_{BP}$ + 10$T_{PM}$ + $T_{PE}$	6$T_{BP}$ + 12$T_{PM}$	4$T_{BP}$ + 8$T_{PM}$ + 2$T_{MTP}$

**Table 3 entropy-24-00648-t003:** Simulation parameters.

Parameters	Values
Simulation area	2.6 km × 2.2 km
Data Transmission Rate	6 Mbps
Transmission Power	20 mW
Noise Floor	−89 dBm
BSM Interval	1 s
Simulation Time	90 s (Simulation 1)/100 s (Simulation 2)
Number of Cars (Simulation 1)	5 (Scenarios 1, 3), 25 (Scenarios 2, 4)
Number of Cars (Simulation 2)	50, 100, 150

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
