# Peer review of "PCP: A Pseudonym Change Scheme for Location Privacy Preserving in VANETs"

_entropy, 2022, doi:10.3390/e24050648_

Round 1

Reviewer 1 Report

The authors have addressed my previous comments.

Minor comment: some figures (12, 14) are very small.

Author Response

We would like to sincerely thank you for your advices and constructive comments. We enlarged the figure 12 and 14 to ensure that readers can clearly understand the content in the figure 12 and 14.

Reviewer 2 Report

The authors need to clarify the following:

1- Can the tracker match vehicles by the beacon message when it has found that the pseudonym is changed. Please explain this in details.

2- How many pseudonyms each vehicle must change in this scheme, please clarify this as it shows that every vehicle must hold lots of pseudonyms to change

3- The conclusion is very short, authors need to elaborate more and connect it to the results.

4- Some references are very old ( e.g. 2009), authors are encouraged to add the following references since they are relevant to the study.

A- Muath Obaidat, I. Shahwan, A. Hassebo, S.  Obeidat, M. Ali, M. Khodjaeva,” SNR-Based Early Warning Message Scheme for VANETs.” Journal of Mobile Multimedia, River Publishers, PP 162-190. Feb. 2020.
B- Muath A. Obaidat, Matluba Khodjaeva, Jennifer Holst, Mohammed Bin Zaid, “Security and Privacy Challenges in Vehicular Networks” Connected Vehicles in the Internet of Things: Concepts, Technologies, and Frameworks for the IoV. Springer, Cham, Switzerland. Feb. 2020.
5- Authors are encouraged to compare their work with the work in "A Blockchain enabled location-privacy preserving scheme for vehicular ad-hoc networks” as they have shown significant improvement. The citation  is below.
Chaudhary, B., Singh, K. A Blockchain enabled location-privacy preserving scheme for vehicular ad-hoc networks. Peer-to-Peer Netw. Appl. 14, 3198–3212 (2021). https://doi.org/10.1007/s12083-021-01079-5.

6- Based on the encryption keys exchange and adopted in  this study, does the scheme prevent DoS/DDoS attack? Please explain this in details.

7- What is the effect of the scheme  on the overall end-to -end delay. As we know delay in E2E delay must be very minimal in VANETs. Please clarify this.

8- The authors assume  that λ = 0.1 , I am wondering what is the effect of changing this value to 0.2, 0.5 and 0.8 ( as similar previous studies) on the scheme’s performance

9- What is the effectiveness of the proposed mechanism in terms of the tracking probability as was introduced in K. Sampigethaya, M. Li, and L. Huang, “AMOEBA: robust location privacy scheme for vanet,” IEEE Journal of Selected Area in Communications, vol. 25, no. 8, pp. 1569–1589, 2016. 

Round 2

Reviewer 2 Report

I just read the new manuscript and followed the changes the authors made. My decision is to accept it in its current form.

This manuscript is a resubmission of an earlier submission. The following is a list of the peer review reports and author responses from that submission.

Round 1

Reviewer 1 Report

This paper proposes a pseudonym change protocol for location privacy-preserving (PCP) in Vehicular Ad Hoc Networks (VANETs). The goal is to provide vehicle anonymity. This new mechanism is optimized for scenarios without Road Side Units (RSU) with low or high vehicle density.

In my opinion, the paper is not technically sound and more work is needed in order to be published in a journal. It is very difficult to understand what are the novelty of the contributions made. A lot of paragraphs only include mathematical notation and formulas without explanations and it makes these sections difficult to read.

My main concerns about this paper are:

  1. As authors say, in the literature, there are several previous proposals that deal with pseudonym change scheme in VANETs. They talk about proposals based on: “silent period”, “mix-zone”, “mix-context”, etc…Besides, these proposals have to work under four type of scenarios, the ones explained in page number 7.

In the paper, there is no a systematic review about these previous works, considering their application in the four types of scenarios. Section 2 only includes a list of works related with mix-zone and mix-content based strategies. However, the proposed scheme is compared, in section 6, with mix-zone based strategies and silent period strategies.

Section 2 should finish with a conclusion section in order to know what are the problems or limitations of the previous schemes and why a new one is needed.

  1. Section 3, about preliminaries, is not well justified. Why are these definitions needed? What is the use of Probabilistic Polynomial Time (PPT) algorithm? The nomenclature used in this section is not explained.
  1. The scheme proposed uses a registration protocol, a V2I authentication protocol, a pseudonym issuance protocol and a pseudonym change protocol. I recommend including the pseudo code for all of them and timelines to show the messages interchanged between the different entities.
  1. The security analysis section might be included as an annex.
  1. In section 6 more detail about the implementation of the scheme is needed. There is no information about how the different protocols of the anonymity scheme are implemented in a network stack for VANETs.
  1. The performance evaluation is only made in a real map scenario and so it is very difficult to validate the results. My recommendation is to implement and analyse the results using the 4 scenarios included in figure 4, besides the real scenario.
  1. Finally, I would have liked a discussion section about implementation and deployment issues in real scenarios.